# PRICING WITH CONTEXTUAL ELASTICITY AND HETEROSCEDASTIC VALUATION

## ABSTRACT

We study an online contextual dynamic pricing problem, where customers decide whether to purchase a product based on its features and price. We introduce a novel approach to modeling a customer's expected demand by incorporating feature-based price elasticity, which can be equivalently represented as a valuation with heteroscedastic noise. To solve the problem, we propose a computationally efficient algorithm called "Pricing with Perturbation (PwP)", which enjoys an $O(\sqrt{dT \log T})$ regret while allowing arbitrary adversarial input context sequences. We also prove a matching lower bound at $\Omega(\sqrt{dT})$ to show the optimality regarding $d$ and $T$ (up to $\log T$ factors). Our results shed light on the relationship between contextual elasticity and heteroscedastic valuation, providing insights for effective and practical pricing strategies.

## 1 INTRODUCTION

Contextual pricing, a.k.a., Feature-based dynamic pricing, considers the problem of setting prices for a sequence of highly specialized or individualized products. With the growth of e-commerce and the increasing popularity of online retailers as well as customers, there has been a growing interest in this area (see, e.g., Amin et al., 2014; Qiang & Bayati, 2016; Javanmard & Nazerzadeh, 2019; Shah et al., 2019; Cohen et al., 2020; Xu & Wang, 2021; Bu et al., 2022).

Formulated as a learning problem, the seller has no prior knowledge of ideal prices but is expected to learn on the fly by exploring different prices and adjusting their pricing strategy after collecting every demand feedback from customers. Different from non-contextual dynamic pricing (Kleinberg & Leighton, 2003) where identical products are sold repeatedly, a contextual pricing agent is expected to generalize from one product to another in order to successfully price a previously-unseen product. A formal problem setup is described below:

---

Contextual pricing. For $t = 1, 2, ..., T$:
1. A product occurs, described by a context $x_t \in \mathbb{R}^d$.
2. The seller (we) proposes a price $p_t \geq 0$.
3. The customer reveals a demand $0 \leq D_t \leq 1$.
4. The seller gets a reward $r_t = p_t \cdot D_t$.

---

Here $T$ is the time horizon, and the (random) demand $D_t$ is drawn from a distribution determined by context (or feature) $x_t$ and price $p_t$. The sequence of contexts $\{x_t\}$ can be either independently and identically distributed (iids) or chosen arbitrarily by an adversary. The seller's goal is to minimize the cumulative *regret* against the sequence of optimal prices.

Existing works on contextual pricing usually assumes linearity on the demand, but they fall into two camps. On the one hand, the "linear demand" camp (Qiang & Bayati, 2016; Ban & Keskin, 2021; Bu et al., 2022) assumes the *demand* $D_t$ as a (generalized) linear model. A typical model is $D_t = \lambda(\alpha p_t + x_t^T \beta) + \epsilon_t$. Here $\alpha < 0$ is a parameter closely related to the *price elasticity*. We will rigorously define a price elasticity in Appendix A.1 according to Parkin et al. (2002), where we also show that $\alpha$ is the *coefficient of elasticity*. Besides of $\alpha$, other parameters like $\beta \in \mathbb{R}^d$ captures the base demand of products with feature $x_t$, $\epsilon_t$ is a zero-mean demand noise, and $\lambda$ is a known monotonically increasing link function. With this model, we have a noisy observation on the expected demand, which is reasonable as the same product is offered many times in period $t$. On the other hand, the "linear valuation" camp (Cohen et al., 2020; Javanmard & Nazerzadeh, 2019; Xu & Wang, 2021) models a buyer's *valuation* $y_t$ as linear and assumes a binary demand $D_t = \mathbb{1}[p_t \leq y_t]$. All

three works listed above assume a *linear-and-noisy* model with $y_t = x_t^\top \theta^* + N_t$, where $\theta^* \in \mathbb{R}^d$ is an unknown linear parameter that captures common valuations and $N_t$ is an idiosyncratic noise assumed to be iid.

Interestingly, the seemingly different modeling principles are closely connected to each other. In the "linear valuation" camp, notice that a customer's probability of "buying" a product equals $\mathbb{E}[D_t]$, which is further given by

$$\mathbb{E}[D_t|p] = \mathbb{P}[y_t \geq p] := S(p - x_t^\top \theta^*),$$

where $S$ is the survival function of $N_t$ (i.e. $S(w) = 1 - \text{CDF}(w)$ for $w \in \mathbb{R}$). This recovers a typical linear demand model by taking $\lambda(w) = S(-w)$ with $\alpha = -1$ and $\beta = \theta^*$. In other words, the distribution of $N_t$ completely characterizes the demand function $\lambda(\cdot)$ and vice versa.

However, the "linear demand" camp is not satisfied with a fixed $\alpha = -1$, while the "linear valuation" camp are skeptical about an observable demand $D_t$ even with zero-mean iid noise. One common limitation to both models is that neither captures how feature $x_t$ affects the price elasticity.

**Our model.** To address this issue, we propose a natural model that unifies the perspectives of both groups. Also, we resolve the common limitation by modeling *heteroscedasticity*, where we assume that the elasticity coefficient $\alpha$ is linearly dependent on feature $x_t$. In specific, we assume:

$$D_t \sim \text{Ber}(S(x_t^\top \eta^* \cdot p_t - x_t^\top \theta^*)), \tag{1}$$

which adopts a generalized linear demand model (GLM) and a Boolean-censored feedback simultaneously. From the perspective of valuation model, it is *equivalent* to assume

$$D_t = \mathbb{1}[p_t \leq y_t], \text{ where } y_t = \frac{1}{x_t^\top \eta^*} \cdot (x_t^\top \theta^* + N_t) \text{ and } \text{CDF}_{N_t}(w) = 1 - S(w). \tag{2}$$

Although Eq. (1) seems more natural than Eq. (2), they are equivalent to each other (with reasonable assumptions on $S$). Notice that the random valuation $y_t$ is *heteroscedastic*, which means its variance is not the same constant across a variety of $x_t$'s. We provide a detailed interpretation of this linear fractional valuation model in appendix.

## 1.1 CONTRIBUTIONS.

Our main results are twofold.

1. We propose a new demand model that assumes a feature-dependent price elasticity on every product. Equivalently, we model the heteroscedasticity on customers' valuations among different products. This model unifies the "linear demand" and "linear valuation" camps.

2. We propose a "Pricing with Perturbation (PwP)" algorithm that achieves $O(\sqrt{dT \log T})$ regret on this model, which is optimal up to $\log T$ factors. This regret upper bound holds for both iid and adversarial $\{x_t\}$ sequences.

## 1.2 TECHNICAL NOVELTY

To the best of our knowledge, we are the first to study a contextual pricing problem with heteroscedastic valuation and Boolean-censored feedback. Some existing works, including Javanmard & Nazerzadeh (2019); Miao et al. (2019); Xu & Wang (2021); Ban & Keskin (2021), focus on related topics and achieve theoretical guarantees. However, their methodologies are not applicable to our settings due to substantial obstacles, which we propose novel techniques to overcome.

**Randomized surrogate regret**. Xu & Wang (2021) solves the problem with $x_t^\top \eta^* = 1$, by taking the negative log-likelihood as a surrogate regret and running an optimization oracle that achieves a fast rate (i.e. an $O(\log T)$ regret). However, the log-likelihood is no longer a surrogate regret in our setting, since it is not "convex enough" and therefore cannot provide sufficient (Fisher) information. In this work, we overcome this challenge by constructing a *randomized* surrogate loss function, whose *expectation* is "strongly convex" enough to upper bound the regret.

**OCO for adversarial inputs**. Javanmard & Nazerzadeh (2019) and Ban & Keskin (2021) study the problem with unknown or heterogeneous noise variances (i.e. elasticity coefficients), but their

techniques highly rely on the distribution of the feature distributions. As a result, their algorithm could be easily attacked by an adversarial $\{x_t\}$ series. In our work, we settle this issue by conducting an online convex optimization (OCO) scheme while updating parameters. Instead of estimating from the history that requires sufficient randomness in the inputs, our algorithm can still work well for adversarial inputs.

In addition, our algorithm has more advanced properties such as computational efficiency and information-theoretical optimality. For more highlights of our algorithm, please refer to Section 4.1.

## 2 RELATED WORKS

Here we present a review of the pertinent literature on contextual pricing and heteroscedasticity in machine learning, aiming to position our work within the context of related studies. For more related works on non-contextual pricing, contextual pricing, contextual searching and contextual bandits, please refer to Wang et al. (2021), Xu & Wang (2021), Krishnamurthy et al. (2021) and Zhou (2015) respectively.

**Contextual Pricing.** As we mentioned in Section 1.2, there are a large number of recent works on contextual dynamic pricing problems, and we refer to Ban & Keskin (2021) as a detailed introduction. On the one hand, Qiang & Bayati (2016); Nambiar et al. (2019); Miao et al. (2019); Ban & Keskin (2021); Bu et al. (2022) assume a (generalized) linear demand model with noise, i.e. $\mathbb{E}[D_t] = g(\alpha p_t - \beta^\top x_t)$. Among those works, Miao et al. (2019) works on a similar setting with a fixed $\alpha$. Ban & Keskin (2021) also assumes a generalized linear demand model, but with independent noises added on demands directly. In comparison, we model $\alpha$ as context-dependent, and we assume the noises are added to valuations instead of demands. Besides, all of those works assume the context sequence $\{x_t\}$ to be iid, whereas we consider it "too good to be true" and work towards an algorithm adaptive to adversarial input sequences. On the other hand, Golrezaei et al. (2019); Shah et al. (2019); Cohen et al. (2020); Javanmard & Nazerzadeh (2019); Xu & Wang (2021); Fan et al. (2021); Goyal & Perivier (2021); Luo et al. (2022) adopts the linear valuation model $y_t = x_t^\top \theta^* + N_t$, which is a special case of our model as $x_t^\top \eta^* = 1$. Specifically, both Javanmard & Nazerzadeh (2019) and Xu & Wang (2021) achieve an $O(d \log T)$ regret with $N_t$ drawn from a known distribution with $x_t^\top \eta^* = -1$. Javanmard & Nazerzadeh (2019) also studies the setting when $x_t^\top \eta^*$ is fixed but unknown and achieves $O(d\sqrt{T})$ regret for stochastic $\{x_t\}$ sequences. In comparison, we achieve $O(\sqrt{dT \log T})$ on a more general problem and get rid of those assumptions.

**Heteroscedasticity.** Since the valuation noise is scaled by a $\frac{1}{x_t^\top \eta^*}$ coefficient, the valuation is *heteroscedastic*, referring to a situation where the variance is not the same constant across all observations. Heteroscedasticity may lead to bias estimates or loss of sample information. There are several existing methods handling this problem, including weighted least squares method (Cunia, 1964), White's test (White, 1980) and Breusch-Pagan test (Breusch & Pagan, 1979). Furthermore, Anava & Mannor (2016) and Chaudhuri et al. (2017) study online learning problems with heteroscedastic variances and provide regret bounds. For a formal and detailed introduction, we refer the audience to the textbook of Kaufman (2013).

## 3 PROBLEM SETUP

### 3.1 NOTATIONS

To formulate the problem, we firstly introduce necessary notations and symbols used in the following sections. The sales session contains $T$ rounds with $T$ known to the seller in advance[1]. At each time $t = 1, 2, \ldots, T$, a product with feature $x_t \in \mathbb{R}^d$ occurs and we propose a price $p_t \geq 0$. Then the nature draws a demand $D_t \sim \text{Ber}(S(x_t^\top \eta^* \cdot p_t - x_t^\top \theta^*))$, where $\theta^*, \eta^* \in \mathbb{R}^d$ are fixed unknown linear parameters and the link function $S : \mathbb{R} \to [0, 1]$ is non-increasing. By the end of time $t$, we receive a reward $r_t = p_t \cdot D_t$.

Equivalently, this customer has a valuation $y_t = \frac{x_t^\top \theta^* + N_t}{x_t^\top \eta^*}$ with noise $N_t \in \mathbb{R}$, and then make a decision $\mathbb{1}_t = \mathbb{1}[p_t \leq y_t] = D_t$ after seeing the price $p_t$. Similarly, we receive a reward $r_t = p_t \cdot \mathbb{1}_t$.

---

[1]Here we assume $T$ known for simplicity. For unknown $T$, we may apply a "doubling epoch" trick as Javanmard & Nazerzadeh (2019) without affecting the regret rate.

Assume $N_t \sim \mathbb{D}_F$ is independently and identically distributed (iid), with cumulative distribution function (CDF) $F = 1 - S$. Denote $s := S'$ and $f := F'$.

## 3.2 DEFINITIONS

Here we define some key quantities. Firstly, we define an expected reward function.

**Definition 3.1** (expected reward function). Define

$$r(u, \beta, p) := \mathbb{E}[r_t | x_t^\top \theta^* = u, x_t^\top \eta^* = \beta, p_t = p] = p \cdot S(\beta \cdot p - u) \tag{3}$$

as the expected reward function.

Given this, we further define a greedy price function as the argmax of $r(u, \beta, p)$ over price $p$.

**Definition 3.2** (greedy price function). Define $J(u, \beta)$ as a greedy price function, i.e. the price that maximizes the expected reward given $u = x_t^\top \theta^*$ and $\beta = x_t^\top \eta^*$.

$$J(u, \beta) = \operatorname*{argmax}_{p \in \mathbb{R}} r(u, \beta, p) = \operatorname*{argmax}_{p \in \mathbb{R}} p \cdot S(\beta \cdot p - u) \tag{4}$$

Notice that

$$J(u, \beta) = \operatorname*{argmax}_{p} p \cdot S(\beta p - u) = \frac{1}{\beta} \cdot \operatorname*{argmax}_{\beta p} \beta p \cdot S(\beta p - u) = \frac{1}{\beta} J(u, 1). \tag{5}$$

According to Xu & Wang (2021, Section B.1), we have the following properties.

**Lemma 3.3.** *Denote* $\varphi(w) := -\frac{S(w)}{s(w)} - w = \frac{1 - F(w)}{f(w)} - w$, *and we have* $J(u, \beta) = \frac{u + \varphi^{-1}(u)}{\beta}$. *Also, for* $u \geq 0$ *and* $\beta > 0$, *we have* $\frac{\partial J(u, \beta)}{\partial u} \in (0, 1)$.

Then we define a negative log-likelihood function of parameter hypothesis $(\theta, \eta)$ given the results at time $t$.

**Definition 3.4** (log-likelihood functions). Denote $\ell_t(\theta, \eta)$ as the negative log-likelihood at time $t$, and define $L_t(\theta, \eta)$ as their summations:

$$-\ell_t(\theta, \eta) = \mathbb{1}_t \cdot \log S(x_t \top \eta \cdot p_t - x_t^\top \theta) + (1 - \mathbb{1}_t) \cdot \log(1 - S(x_t^\top \eta \cdot p_t - x_t^\top \theta)).$$
$$L_t(\theta, \eta) = \sum_{\tau=1}^{t} \ell_t. \tag{6}$$

Finally, we define the round-$t$ expected regret and cumulative expected regret.

**Definition 3.5** (regrets). Define $Reg_t(p_t) := r(x_t^\top \theta^*, x_t^\top \eta^*, J(x_t^\top \theta^*, x_t^\top \eta^*)) - r(x_t^\top \theta^*, x_t^\top \eta^*, p_t)$ as the expected regret at round $t$, conditioning on price $p_t$. Also, define the cumulative regret as follows

$$Regret = \sum_{t=1}^{T} Reg_t(p_t) \tag{7}$$

## 3.3 ASSUMPTIONS

We establish three technical assumptions to make our analysis and presentation clearer. Firstly, we assume that all feature and parameter vectors are bounded within a unit ball in Euclidean norm. This assumption is without loss of generality as it only rescales the problem.

**Assumption 3.6** (bounded feature and parameter spaces). Assume features $x_t \in \mathcal{H}_x$ and parameters $\theta \in \mathcal{H}_\theta, \eta \in \mathcal{H}_\eta$. Denote $U_p^d := \{x \in \mathbb{R}^d, \|x\|_p \leq 1\}$ as an $L_p$-norm unit ball in $\mathbb{R}^d$. Assume all $\mathcal{H}_x, \mathcal{H}_\theta, \mathcal{H}_\eta \in U_p^d$. Also, assume $x^\top \theta > 0, \forall x \in \mathcal{H}_x, \theta \in \mathcal{H}_\theta$ and $x^\top \eta > C_\beta > 0, \forall x \in \mathcal{H}_x, \eta \in \mathcal{H}_\eta$ for some constant $C_\beta$.

We will show the necessity of assuming an elasticity lower bound $C_\beta$ in Appendix C. In specific, we claim that any algorithm will suffer a regret of $\Omega(\frac{1}{C_\beta})$. For the simplicity of notation, we denote $[\theta; \eta] := [\theta^\top, \eta^\top]^\top \in \mathbb{R}^{2d}$ as the combination of $d$-dimension column vectors $\theta$ and $\eta$. Since we know that $x_t^\top \theta \in [0,1]$ and $x_t^\top \eta \in [C_\beta, 1]$, we have $J(x_t^\top \theta, x_t^\top \eta) \in [J(0,1), J(1, C_\beta)]$. Later we will show that the price perturbation is no more than $\frac{J(0,1)}{10}$. Therefore, we may have the following assumption.

**Assumption 3.7** (bounded prices). For any price $p_t$ at each time $t = 1, 2, \ldots, T$, we require $p_t \in [c_1, c_2]$, where $c_1 = \frac{J(0,1)}{2}$ and $c_2 = 2J(1, C_\beta)$.

Similar to Javanmard & Nazerzadeh (2019) and Xu & Wang (2021), we also assume a log-concavity on the noise CDF.

**Assumption 3.8** (log-concavity). Every $D_t$ is independently sampled according to Eq. (1), with $S(\omega) \in [0,1]$ and $s(\omega) = S'(\omega) > 0, \forall \omega \in \mathbb{R}$. Equivalently, the valuation noise $N_t \sim \mathbb{D}_F$ is independently and identically distributed (iid), with CDF $F = 1 - S$. Assume that $S \in \mathbb{C}^2$, and $S$ and $(1-S)$ are strictly log-concave.

## 4 MAIN RESULTS

To solve the contextual pricing problem with featurized elasticity, we propose our "Pricing with Perturbation (PwP)" algorithm. In the following, we firstly describe the algorithm and highlight its properties, then analyze (and bound) its cumulative regret, and finally prove a regret lower bound to show its optimality.

### 4.1 ALGORITHM

The pseudocode of PwP is displayed as Algorithm 1, which calls an ONS oracle (Algorithm 2).

---
**Algorithm 1** Pricing with Perturbation (PwP)
---
1: **Input:** parameter spaces $\mathcal{H}_\theta, \mathcal{H}_\eta$, link function $S$, time horizon $T$, dimension $d$
2: **Initialization:** parameters $\theta_1 \in \mathcal{H}_\theta$, $\eta_1 \in \mathcal{H}_\eta$, price perturbation $\Delta$, cumulative likelihood $L_0 = 0$, matrix $A_0 = \epsilon \cdot I_{2d}$ and parameter $\epsilon, \gamma$
3: **for** $t = 1, 2, \ldots, T$ **do**
4:     Observe $x_t$;
5:     Calculate greedy price $\hat{p}_t = J(x_t^\top \theta_t, x_t^\top \eta_t)$
6:     Sample $\Delta_t = \Delta$ with $\Pr = 0.5$ and $\Delta_t = -\Delta$ with $\Pr = 0.5$;
7:     Propose price $p_t = \hat{p}_t + \Delta_t$;
8:     Receive the customer's decision $\mathbb{1}_t$;
9:     Construct negative log-likelihood $\ell_t(\theta, \eta)$ and $L_t(\theta, \eta)$ as eq. (6);
10:     Update parameters:
$$[\theta_{t+1}; \eta_{t+1}] \leftarrow ONS([\theta_t; \eta_t])$$
11: **end for**

---

At each time $t$, it inherits parameters $\theta_t$ and $\eta_t$ from $(t-1)$ and takes in a context vector $x_t$. By trusting in $\theta_t$ and $\eta_t$, it calculates a greedy price $\hat{p}_t$ and outputs a perturbed version $p_t = \hat{p}_t + \Delta_t$. After seeing customer's decision $\mathbb{1}_t$, PwP calls an "Online Newton Step (ONS)" oracle (see Algorithm 2) to update the parameters as $\theta_{t+1}$ and $\eta_{t+1}$ for future use.

#### 4.1.1 HIGHLIGHTS

We highlight the achievements of the PwP algorithm in the following three aspects.

**In this pricing problem.** As we mentioned in Section 1.2, the key to solving this contextual elasticity (or heteroscedastic valuation) pricing problem is to construct a surrogate loss function. Xu & Wang (2021) adopts negative log-likelihood in their setting, which does not work for ours since it is not "convex" enough. In our PwP algorithm, we overcome this challenge by introducing a perturbation $\Delta$ on the proposed greedy price. This idea originates from the observation that the *variance* of $p_t$ contributes positively to the "convexity" of the expected log-likelihood, which helps "re-build" the upper-bound inequality.

---

**Algorithm 2** Online Newton Step (ONS)

---

1: **Input: current parameter $[\theta_t, \eta_t]$, likelihood $\ell_t(\theta, \eta)$, matrix $A_t$, parameter $\gamma$, parameter spaces $\mathcal{H}_\theta$ and $\mathcal{H}_\eta$.**
2: Calculate $\nabla_t = \nabla \ell_t(\theta, \eta)$;
3: Rank-1 update: $A_t = A_{t-1} + \nabla_t \nabla_t^\top$;
4: Newton step: $[\hat{\theta}_{t+1}; \hat{\eta}_{t+1}] = [\hat{\theta}_t; \hat{\eta}_t] - \frac{1}{\gamma} A_t^{-1} \nabla_t$;
5: Projection: $[\theta_{t+1}; \eta_{t+1}] = \Pi_{\mathcal{H}_\theta \times \mathcal{H}_\eta}^{A_t}([\hat{\theta}_{t+1}; \hat{\eta}_{t+1}])$;

---

**In online optimization.** PwP perturbs the action (price) it should have taken greedily. This idea is similar to a "Following the Perturbed Leader (FTPL)" algorithm (Hutter et al., 2005) that minimizes the summation of the empirical risk and a random loss function serving as a perturbation. However, this might lead to extra computational cost as the random perturbation is not necessarily smooth and therefore hard to optimize. In this work, PwP introduces a possible way to overcome this obstacle: Instead of perturbing the objective function, we may directly perturb the action to explore its neighborhood. Our regret analysis and results indicate the optimality of this method and imply a potentially wide application.

**In information theory.** In the regret analysis of PwP, we show that: By adding $\Delta$ perturbation on $p_t$, we may lose $O(\Delta^2)$ in reward but will gain $O(\Delta^2) \cdot I$ in Fisher information (i.e. the expected Hessian of negative log-likelihood function) in return. By Cramer-Rao Bound, this leads to $O(\frac{1}{\Delta^2})$ estimation error. In this way, we quantify the information (observing from exploration) on the scale of reward, which shares the same idea with the Upper Confidence Bound (Lai & Robbins, 1985) method that always maximizes the summation of empirical reward and information-traded reward.

Besides, PwP is computationally efficient as it only calls the ONS oracle for once. As for the ONS oracle, it updates an $A_t^{-1} = (A_{t-1} + \nabla_t \nabla_t^\top)^{-1}$ at each time $t$, which is with $O(d^2)$ time complexity according to the following *Woodbury matrix identity*

$$(A + xx^\top)^{-1} = A^{-1} - \frac{1}{1 + x^\top A^{-1} x} A^{-1} x (A^{-1} x)^\top. \tag{8}$$

## 4.2 REGRET UPPER BOUND

Now we analyze the regret of PwP and propose an upper bound up to constant coefficients.

**Theorem 4.1.** *Under Assumption 3.6, 3.7 and 3.8, by taking $\Delta = \min\left\{ \left(\frac{d \log T}{T}\right)^{\frac{1}{4}}, \frac{J(0,1)}{10}, \frac{1}{10} \right\}$, the algorithm PwP guarantees an expected regret at $O(\sqrt{dT \log T})$.*

In the following, we prove Theorem 4.1 by stating a thread of key lemmas. We leave the detailed proof of those lemmas to Appendix A.

*Proof.* The proof overview can be displayed as the following roadmap of inequalities:

$$\mathbb{E}[Regret] = \sum_{t=1}^T Reg_t(p_t) \leq \mathbb{E}\left[\sum_{t=1}^T O\left((x_t^\top(\theta_t - \theta^*))^2 + (x_t^\top(\eta_t - \eta^*))^2 + \Delta^2\right)\right]$$

$$\leq O\left(\frac{\sum_{t=1}^T \mathbb{E}\left[\ell_t(\theta_t, \eta_t) - \ell_t(\theta^*, \eta^*)\right]}{\Delta^2} + T \cdot \Delta^2\right) \tag{9}$$

$$\leq O\left(\frac{d \log T}{\Delta^2} + T \cdot \Delta^2\right) = O(\sqrt{dT \log T}).$$

Here the first inequality is by the smoothness of regret function (see Lemma 4.2), the second inequality is by a special "strong convexity" of $\ell_t(\theta, \eta)$ that contributes to the surrogate loss (see Lemma 4.3), the third inequality is by Online Newton Step (see Lemma 4.4), and the last equality is by the value of $\Delta$. A rigorous version of Eq. (9) can be found in Appendix A.4.

We firstly show the smoothness of $Reg_t(p_t)$:

**Lemma 4.2** (regret smoothness). *Denote $p_t^* := J(x_t^\top \theta^*, x_t^\top \eta^*)$. There exists constants $C_r > 0$ and $C_J > 0$ such that*

$$Reg_t(p_t) \leq C_r \cdot (p_t - p_t^*)^2 \leq C_r \cdot 2 \left( C_J \cdot \left[ (x_t^\top(\theta_t - \theta^*))^2 + (x_t^\top(\eta_t - \eta^*))^2 \right] + \Delta^2 \right). \quad (10)$$

While the first inequality of Eq. (10) is from the smoothness, and the second inequality is by the Lipschitzness of function $J(u, \beta)$. Please refer to Appendix A.2 for proof details. We then show the reason why the log-likelihood function can still be a surrogate loss with carefully randomized $p_t$.

**Lemma 4.3** (surrogate expected regret). *There exists a constant $C_l > 0$ such that $\forall \theta \in \mathcal{H}_\theta, \eta \in \mathcal{H}_\eta$, we have*

$$\mathbb{E}[\ell_t(\theta, \eta) - \ell_t(\theta^*, \eta^*)|\theta_t, \eta_t]$$
$$\geq \frac{C_l \Delta^2}{10}[(\theta - \theta^*)^\top, (\eta - \eta^*)^\top] \begin{bmatrix} x_t x_t^\top & 0 \\ 0 & x_t x_t^\top \end{bmatrix} \begin{bmatrix} \theta - \theta^* \\ \eta - \eta^* \end{bmatrix} \quad (11)$$
$$= \frac{C_l \cdot \Delta^2}{10} \left[ \left(x_t^\top(\theta - \theta^*)\right)^2 + \left(x_t^\top(\eta - \eta^*)\right)^2 \right].$$

This is the most important lemma in this work. We show a proof sketch here and defer the detailed proof to Appendix A.3.

*Proof sketch of Lemma 4.3.* We show that there exist constants $C_l > 0, C_p > 0$ such that

1. $\nabla^2 \ell_t(\theta, \eta) \succeq C_l \cdot \begin{bmatrix} x_t x_t^\top & -p_t \cdot x_t x_t^\top \\ -p_t \cdot x_t x_t^\top & p_t^2 \cdot x_t x_t^\top \end{bmatrix}$, and

2. $\mathbb{E} \begin{bmatrix} x_t x_t^\top & -p_t \cdot x_t x_t^\top \\ -p_t \cdot x_t x_t^\top & p_t^2 \cdot x_t x_t^\top \end{bmatrix} |\theta_t, \eta_t ] \succeq C_p \Delta^2 \begin{bmatrix} x_t x_t^\top & 0 \\ 0 & x_t x_t^\top \end{bmatrix}$.

The first property above relies on the exp-concavity of $\ell_t$. Notice that the second property does not hold without the $\mathbb{E}$ notation, as the left hand side is a $(a - b)^2$ form while the right hand side is in a $(a^2 + b^2)$ form. In general, there exist no constant $c > 0$ such that $(a - b)^2 \geq c(a^2 + b^2)$. However, due to the randomness of $p_t$, we have

$$\mathbb{E}[p_t^2|\hat{p}_t] = \mathbb{E}[p_t|\hat{p}_t]^2 + \Delta^2. \quad (12)$$

In this way, the *conditional expectation* of the left hand side turns to $(a - b)^2 + \lambda \cdot b^2$ and we have

$$(a - b)^2 + \lambda b^2 = (\frac{1}{\sqrt{1 + \frac{\lambda}{2}}} \cdot a - \sqrt{1 + \frac{\lambda}{2}} \cdot b)^2 + (1 - \frac{1}{1 + \frac{\lambda}{2}})a^2 + \frac{\lambda}{2}b^2 \geq \frac{\frac{\lambda}{2}}{1 + \frac{\lambda}{2}} \cdot (a^2 + b^2).$$
$$(13)$$

Similarly, we upper bound $\begin{bmatrix} x_t x_t^\top & 0 \\ 0 & x_t x_t^\top \end{bmatrix}$ with $\mathbb{E}[\nabla^2 \ell_t(\theta, \eta)|\theta_t, \eta_t]$ up to a $C_p \cdot \Delta^2$ coefficient.

With those two properties above, along with a property of likelihood function that $\mathbb{E}[\nabla \ell_t(\theta^*, \eta^*)] = 0$, we can prove Lemma 4.3 by taking a Taylor expansion of $\ell_t$ at $[\theta^*; \eta^*]$. ∎

Finally, we cite a theorem from Hazan (2016) as our Lemma 4.4 that reveals the surrogate regret rate on negative log-likelihood functions.

**Lemma 4.4.** *With parameters $G = \sup_{\theta \in \mathcal{H}_\theta, \eta \in \mathcal{H}_\eta} \|\nabla l_t(\theta, \eta)\|_2$, $D = \sup \|[\theta_1; \eta_1] - [\theta_2; \theta_2]\| \leq 2$, $\alpha = C_e$, $\gamma = \frac{1}{2} \min\{\frac{1}{4GD}, \alpha\}$ and $\epsilon = \frac{1}{\gamma^2 D^2}$ and $T > 4$, Keep running Algorithm 2 for $t = 1, 2, \ldots, T$ guarantees:*

$$\sup_{\{x_t\}} \left\{ \sum_{t=1}^T \ell_t(\theta_t, \eta_t) - \min_{\theta \in \mathcal{H}_\theta, \eta \in \mathcal{H}_\eta} \sum_{t=1}^T \ell_t(\theta, \eta) \right\} \leq 5(\frac{1}{\alpha} + GD)d \log T. \quad (14)$$

With all these lemma above, we have proved every line of Eq. (9). ∎

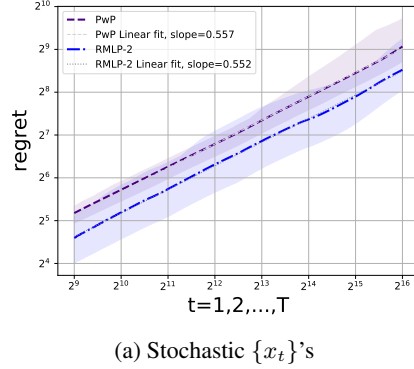
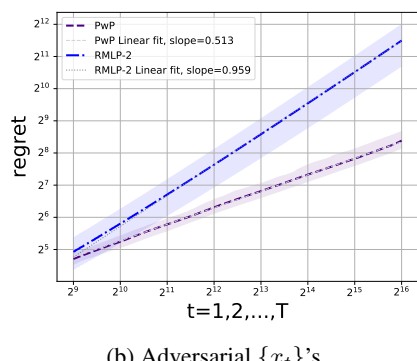

(a) Stochastic $\{x_t\}$'s

(b) Adversarial $\{x_t\}$'s

Figure 1: The regret of PwP algorithm and a modified RMLP-2 algorithm on simulation data (generated according to Eq. (1)), plotted in log-log scales to indicate the regret dependence on $T$. Figure 1a and Figure 1b are for stochastic and adversarial $\{x_t\}$ sequences respectively. We also plot linear fits for those regret curves, where a slope-$\alpha$ line indicates an $O(T^\alpha)$ regret. The error bands are drawn with 0.95 coverage using Wald's test. From the figures, we know that PwP performs closely to its $O(\sqrt{T \log T})$ regret regardless of the types of input context sequences, whereas RMLP-2 fails in the attack of adversarial input.

### 4.3 LOWER BOUNDS

We claim that PwP is near-optimal in information theory, by proposing a matching regret lower bound in Theorem 4.5. We present the proof with valuation model to match with existing results.

**Theorem 4.5.** *Consider the contextual pricing problem setting with Bernoulli demand model given in Eq. (1). With all assumptions in Section 3 hold, any pricing algorithm has to suffer a $\Omega(\sqrt{dT})$ worst-case regret, with $T$ the time horizon and $d$ the dimension of context.*

*Proof.* The main idea is to reduce $d$ numbers of 1-dimension problems to this problem setting. In fact, we may consider the following problem setting:

1. Construct set $X = \{x_i := [0, \ldots, 0, 1, 0, \ldots, 0]^\top \in \mathbb{R}^d$ with only $i^{\text{th}}$ place being $1, i = 1, 2, \ldots, d\}$.

2. Let $\theta^* = [\frac{u_1}{\sigma_1}, \frac{u_2}{\sigma_2}, \frac{u_3}{\sigma_3}, \ldots, \frac{u_d}{\sigma_d}]^\top, \eta^* = [\frac{1}{\sigma_1}, \frac{1}{\sigma_2}, \frac{1}{\sigma_3}, \ldots, \frac{1}{\sigma_d}]^\top$, and therefore we have $\frac{x_i^\top \theta^* + N_t}{x_i^\top \eta^*} = u_i + \sigma_i \cdot N_t$.

3. At each time $t = 1, 2, \ldots, T$, sample $x_t \sim X$ independently and uniformly at random.

In this way, we divide the whole time series $T$ into $d$ separated sub-problems: $y_t(i) = u_i + \sigma_i \cdot N_t$. Notice that we know the distribution of $N_t$, any feedback on a round $t$ with feature $x_i$ would not provide any information on another round $t'$ with feature $x_j$ if $i \neq j$. For each sub-problem, it has a time horizon as $T/d$ in expectation. According to Broder & Rusmevichientong (2012, Theorem 3.1), the regret lower bound of each sub-problem is $\Omega(\sqrt{\frac{T}{d}})$. Therefore, the total regret lower bound is $d \cdot \sqrt{\frac{T}{d}} = \sqrt{Td}$. ■

## 5 NUMERICAL EXPERIMENTS

Here we conduct numerical experiments to validate the low-regret performance of our algorithm PwP. Since we are the first to study this heteroscadestic valuation model, we do not have a baseline algorithm working for exactly the same problem. However, we can modify the RMLP-2 algorithm in Javanmard & Nazerzadeh (2019) by only replacing their max-likelihood estimator (MLE) for $\theta^*$ with a new MLE for both $\theta^*$ and $\eta^*$. This modified RMLP-2 algorithm does not have a regret guarantee in our setting, but it may still serve as a baseline to compare with.

We test PwP and the modified RMLP-2 on the demand model assumed in Eq. (1) with both stochastic and adversarial $\{x_t\}$ sequences, respectively. Basically, we assume $T = 2^{16}$ $d = 2$, $N_t \sim \mathcal{N}(0, \sigma^2)$ with $\sigma = 0.5$, and we repeatedly run each algorithm for 20 times in each experiment setting. In order

to show the regret dependence w.r.t. $T$, we plot all cumulative regret curves in log-log plots, where an $\alpha$ slope indicates an $O(T^\alpha)$ dependence.

**Stochastic $\{x_t\}$.** We implement and test PwP and RMLP-2 on stochastic $\{x_t\}$'s, where $x_t$ are iid sampled from $\mathcal{N}(\mu_x, \Sigma_x)$ (for $\mu_x = [10, 10, \ldots, 10]^\top$ and some randomly sampled $\Sigma_x$) and then normalized s.t. $\|x_t\|_2 \le 1$. The numerical results are shown in Figure 1a. Numerical results show that both algorithms achieve $\sim O(T^{0.56})$ regrets, which is close to the theoretic regret rate at $O(\sqrt{T \log T})$.

**Adversarial $\{x_t\}$.** Here we design an adversarial $\{x_t\}$ sequence to attack both algorithms. Since RMLP-2 divides the whole time horizon $T$ into epochs with length $k = 1, 2, 3, \ldots$ sequentially and then does pure exploration at the beginning of each epoch, we may directly attack those pure-exploration rounds in the following way: (1) In each pure-exploration round (i.e. when $t = 1, 3, 6, \ldots, \frac{k(k+1)}{2}, \ldots$), let the context be $x_t = [1, 0]^\top$; (2) In any other round, let the context be $x_t = [0, 1]^\top$. In this way, the RMLP-2 algorithm will never learn $\theta^*[2]$ and $\eta^*[2]$ since the inputs of pure-exploration rounds do not contain this information. Under this oblivious adversarial context sequence, we implement PwP and RMLP-2 and compare their performance. The results are shown in Figure 1b, indicating that PwP can still guarantee $O(T^{0.513})$ regret (close to $O(\sqrt{T \log T})$) while RMLP-2 runs into a linear regret.

As a high-level interpretation, the performance difference is because PwP adopts a "distributed" exploration at every time $t$ while RMLP-2 makes it more "concentrated". Although both PwP and RMLP-2 take the same amount of exploration that optimally balance the reward loss and the information gain (and that is why they both perform well in stochastic inputs), randomly distributed exploration would save the algorithm from being "attacked" by oblivious adversary. In fact, this phenomenon is analog to $\epsilon$-Greedy versus Exploration-first algorithms in multi-armed bandits. We will discuss more in Appendix C.

So far, we have presented the numerical results of running PwP and a modified RMLP-2 on the well-assumed demand model as Eq. (1) (or Eq. (2) equivalently). Besides of that, we also conduct experiments on a model-misspecification setting to show the robustness, where the true demand (or valuation) distribution is not the same as Eq. (1) or Eq. (2). The numerical results are presented in Appendix B.

## 6 DISCUSSION

Here we discuss the motivation and the limitation of making Assumption 3.6. We leave the majority of discussion to Appendix C.

**Necessity of lower-bounding $x_t^\top \eta^*$ from 0.** As we state in Assumption 3.6, the price elasticity coefficient $x_t^\top \eta^*$ is lower bounded by a constant $C_\beta > 0$. On the one hand, this is necessary since we cannot have an upper bound on the optimal price without this assumption. On the other hand, according to Eq. (3), we know that $r(u, \beta, p) = r(u, 1, \beta \cdot p) \cdot \frac{1}{\beta}$, which indicates that the reward is rescaled by $\frac{1}{\beta}$. As a result, the regret should be proportional to $\frac{1}{C_\beta}$. Although a larger (i.e. closer to 0) elasticity would lead to a more *smooth* demand curve, this actually reduce the information we could gather from customers' feedback and slow down the learning process. We look forward to future researches getting rid of this assumption and achieve more adaptive regret rates.

## 7 CONCLUSION

In summary, our work focuses on the problem of contextual pricing with highly differentiated products. We propose a contextual elasticity model that unifies the "linear demand" and "linear valuation" camps and captures the price effect and heteroscedasticity. To solve this problem, we develop an algorithm PwP, which utilizes Online Newton Step (ONS) on a surrogate loss function and proposes perturbed prices for exploration. Our analysis show that it guarantees a $O(\sqrt{dT \log T})$ regret even for adversarial context sequences. We also provide a matching $\Omega(\sqrt{dT})$ regret lower bound to show its optimality (up to $\log T$ factors). Besides, our numerical experiments also validate the regret bounds of PwP and its advantage over existing method. We hope this work would shed lights on the research of contextual pricing as well as online decision-making problems.

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

# A DEFINITION AND PROOF DETAILS

Here we show the proof details of the lemmas we have stated in Section 4.2. Before that, let us clarify some terminologies we mentioned in the main paper.

## A.1 DEFINITIONS

Firstly, we rigorously define the concept of *price elasticity* occurring in Section 1.

**Definition A.1** (Price Elasticity (Parkin et al., 2002)). Suppose $D(p)$ is a demand function of price $p$. Then the price elasticity $E_d$ of demand is defined as

$$E_D := \frac{\Delta D(p)/D(p)}{\Delta p/p} = \frac{\partial D(p)}{\partial p} \cdot \frac{p}{D(p)}. \tag{15}$$

With this definition, along with our generalized linear demand model given in Eq. (1), the price elasticity for the expected demand $S(x_t^\top \eta^* \cdot p_t - x_t^\top \theta^*)$ is

$$
\begin{aligned}
E_D =& \frac{\partial S(x_t^\top \eta^* \cdot p_t - x_t^\top \theta^*)}{\partial p_t} \cdot \frac{p_t}{S(x_t^\top \eta^* \cdot p_t - x_t^\top \theta^*)} \\
=& x_t^\top \eta^* \cdot \frac{s(x_t^\top \eta^* \cdot p_t - x_t^\top \theta^*)}{S(x_t^\top \eta^* \cdot p_t - x_t^\top \theta^*)} \cdot p_t.
\end{aligned} \tag{16}
$$

Here $s(\cdot) = S'(\cdot)$. Therefore, despite the effect of the link function and the price $p_t$, the price elasticity is proportional to the price coefficient $x_t^\top \eta^*$. This is why we call $x_t^\top \eta^*$ (or $\alpha$ in the general model $D(p) = \lambda(\alpha \cdot p + x_t^T \beta)$) the *elasticity coefficient* or *coefficient of elasticity* in Section 1.

## A.2 PROOF OF LEMMA 4.2

*Proof.* In order to prove Lemma 4.2, we show the following lemma that indicates the Lipschitzness of $J(u, \beta)$:

**Lemma A.2** (Lipschitz of optimal price). *There exists a constant $C_J > 0$ such that*

$$|J(u_1, \beta_1) - J(u_2, \beta_2)| \le C_J \cdot (|u_1 - u_2| + |\beta_1 - \beta_2|). \tag{17}$$

With this lemma, we get the second inequality of Eq. (10). We will prove this lemma later in this subsection. Now, we focus on the first inequality. Notice that

$$
\begin{aligned}
Reg_t(p_t) =& r(x_t^\top \theta^*, x_t^\top \eta^*, p_t^*) - r(x_t^\top \theta^*, x_t^\top \eta^*, p_t) \\
\le& -\frac{\partial r(u, \beta, p)}{\partial p}\Big|_{u=x_t^\top \theta^*, \beta=x_t^\top \eta^*, p=p_t^*}(p_t^* - p_t) \\
& -\frac{1}{2} \cdot \inf_{p \in [c_1, c_2], \beta \in [C_\beta, 1], u \in [0,1]} \frac{\partial^2 r(u, \beta, p)}{\partial p^2}\Big|_{u=x_t^\top \theta^*, \beta=x_t^\top \eta^*, p=p_t^*}(p_t^* - p_t)^2 \\
=& 0 + \frac{1}{2} \cdot \sup_{p \in [c_1, c_2], \beta \in [C_\beta, 1], u \in [0,1]} \{|2s(\beta \cdot p - u) \cdot \beta + p \cdot s'(\beta \cdot p - u) \cdot \beta^2|\}(p_t^* - p_t)^2.
\end{aligned} \tag{18}
$$

Here the first line is by the definition of $Reg_t(p_t)$, the second line is by smoothness, the third line is by the optimality of $p_t^*$, and the last line is by calculus. Since $|2s(\beta \cdot p - u) \cdot \beta + p \cdot s'(\beta \cdot p - u) \cdot \beta^2|$ is continuous on $p \in [c_1, c_2], \beta \in [C_\beta, 1], u \in [0, 1]$, we denote this maximum as $2C_r$, which proves the first inequality of Eq. (10). ∎

Now we show the proof of Lemma A.2.

*Proof of Lemma A.2.* Since $J(u, \beta) = \frac{u + \varphi^{-1}(u)}{\beta}$ where $\varphi(w) = -\frac{S(w)}{s(w)} - w$. Notice that

$$\varphi'(w) = -\frac{d\frac{S(w)}{s(w)}}{dw} - 1 = \frac{d^2 \log(S(w))}{dw^2} \cdot \frac{S(w)^2}{s(w)^2} - 1 < -1 \tag{19}$$

since $S(w)$ is log-concave (as is assumed in Assumption 3.8). Given Eq. (19), we know that $\frac{d\varphi^{-1}(u)}{d(u)} = \frac{1}{\frac{d\varphi(w)}{dw}|_{w=\varphi^{-1}(u)}} \in (-1, 0)$. Therefore, we have:

$$\frac{\partial J(u, \beta)}{\partial u} = \frac{1 + \frac{d\varphi^{-1}(u)}{du}}{\beta} \in (0, \frac{1}{C_\beta})$$

$$\frac{\partial J(u, \beta)}{\partial \beta} = \frac{\partial \frac{J(u,1)}{\beta}}{\partial \beta} = -\frac{J(u, 1)}{\beta^2} \in [-\frac{c_2}{C_\beta}, -c_1].$$

(20)

Therefore, we know that $J(u, \beta)$ is Lipschitz with respect to $u$ and $\beta$ respectively. Take $C_J = \max\{\frac{1}{C_\beta}, \frac{c_2}{C_\beta}\}$ and we get Eq. (17). ∎

### A.3  PROOF OF LEMMA 4.3

*Proof.* We firstly show the convexity (and exp-concavity) of $\ell_t(\theta, \eta)$ by the following lemma.

**Lemma A.3** (exp-concavity). *$\ell_t(\theta, \eta)$ is convex and $C_e$-exp-concave with respect to $[\theta; \eta]$, where $C_e > 0$ is a constant dependent on $F$ and $C_\beta$. Equivalently, $\nabla^2 \ell_t(\theta, \eta) \succeq C_e \cdot \nabla \ell_t(\theta, \eta) \nabla \ell_t(\theta, \eta)^\top$. Also, we have $\nabla^2 \ell_t(\theta, \eta) \succeq C_l \cdot \begin{bmatrix} x_t x_t^\top & -p_t \cdot x_t x_t^\top \\ -p_t \cdot x_t x_t^\top & v_t^2 \cdot x_t x_t^\top \end{bmatrix}$ for some constant $C_l > 0$.*

The proof of Lemma A.3 is mainly straightforward calculus, and we defer the proof to the end of this subsection. According to Lemma A.3, we have $\nabla^2 \ell_t(\theta, \eta) \succeq C_l \cdot \begin{bmatrix} x_t x_t^\top & -p_t \cdot x_t x_t^\top \\ -p_t \cdot x_t x_t^\top & p_t^2 \cdot x_t x_t^\top \end{bmatrix}$. Therefore, we know that

$$\ell_t(\theta, \eta) \geq \ell_t(\theta^*, \eta^*) + \nabla \ell_t(\theta^*, \eta^*)^\top \begin{bmatrix} \theta - \theta^* \\ \eta - \eta^* \end{bmatrix} + [(\theta - \theta^*)^\top, (\eta - \eta^*)^\top] C_l \begin{bmatrix} x_t x_t^\top & -p_t x_t x_t^\top \\ -p_t x_t x_t^\top & p_t^2 x_t x_t^\top \end{bmatrix} \begin{bmatrix} \theta - \theta^* \\ \eta - \eta^* \end{bmatrix}$$

(21)

According to the property of likelihood, we have $\mathbb{E}[\nabla \ell_t(\theta^*, \eta^*)|\theta_t, \eta_t] = 0$ for any $\theta_t$ and $\eta_t$. Combining this with Eq. (21), we get

$$\mathbb{E}[\ell_t(\theta, \eta) - \ell_t(\theta^*, \eta^*)|\theta_t, \eta_t] \geq C_l[(\theta - \theta^*)^\top, (\eta - \eta^*)^\top] \mathbb{E} \begin{bmatrix} x_t x_t^\top & -p_t x_t x_t^\top \\ -p_t x_t x_t^\top & p_t^2 x_t x_t^\top \end{bmatrix} |\theta_t, \eta_t] \begin{bmatrix} \theta - \theta^* \\ \eta - \eta^* \end{bmatrix}$$

(22)

Recall that $\hat{p}_t = J(x_t^\top \theta_t, x_t^\top \eta_t)$ and that $p_t = \hat{p}_t + \Delta_t$. Therefore, we know that the conditional expectations $\mathbb{E}[p_t|\theta_t, \eta_t] = \hat{p}_t$ and $\mathbb{E}[p_t^2|\theta_t, \eta_t] = \hat{p}_t^2 + \Delta^2$. Given this, we have

$$\mathbb{E} \begin{bmatrix} x_t x_t^\top & -p_t x_t x_t^\top \\ -p_t x_t x_t^\top & p_t^2 x_t x_t^\top \end{bmatrix} |\theta_t, \eta_t]$$

$$= \begin{bmatrix} x_t x_t^\top & -\hat{p}_t x_t x_t^\top \\ -\hat{p}_t x_t x_t^\top & (\hat{p}_t^2 + \Delta^2) x_t x_t^\top \end{bmatrix}$$

$$= \begin{bmatrix} x_t \\ -\hat{p}_t x_t \end{bmatrix} [x_t^\top, -\hat{p}_t x_t^\top] + \begin{bmatrix} 0 & 0 \\ 0 & \Delta^2 x_t x_t^\top \end{bmatrix}$$

$$= \begin{bmatrix} \frac{1}{\sqrt{1+\frac{\Delta^2}{2}}} \cdot x_t \\ -\sqrt{1 + \frac{\Delta^2}{2}} \hat{p}_t \cdot x_t \end{bmatrix} \begin{bmatrix} \frac{1}{\sqrt{1+\frac{\Delta^2}{2}}} \cdot x_t^\top, -\sqrt{1 + \frac{\Delta^2}{2}} \hat{p}_t \cdot x_t^\top \end{bmatrix} + \begin{bmatrix} (1 - \frac{1}{1+\frac{\Delta^2}{2}}) x_t x_t^\top & 0 \\ 0 & \frac{\Delta^2}{2} x_t x_t^\top \end{bmatrix}$$

(23)

Since $\Delta = \min \left\{ \left( \frac{d \log T}{T} \right)^{\frac{1}{4}}, \frac{J(0,1)}{10}, \frac{1}{10} \right\}$, we have $1 - \frac{1}{1+\frac{\Delta^2}{2}} = \frac{\frac{\Delta^2}{2}}{1+\frac{\Delta^2}{2}} \geq \frac{\Delta^2}{10}$. As a result, we have

$$\mathbb{E} \begin{bmatrix} x_t x_t^\top & -p_t x_t x_t^\top \\ -p_t x_t x_t^\top & p_t^2 x_t x_t^\top \end{bmatrix} |\theta_t, \eta_t]$$

$$\geq \frac{\Delta^2}{10} \cdot \begin{bmatrix} x_t x_t^\top & 0 \\ 0 & x_t x_t^\top \end{bmatrix}$$

(24)

This proves the lemma. ∎

Finally, we show the proof of Lemma A.3.

*Proof of Lemma A.3.* Recall that $\ell_t(\theta, \eta) = -\mathbb{1}_t \cdot \log(S(x_t^\top(p_t\eta - \theta))) - (1 - \mathbb{1}_t) \cdot \log(1 - S(x_t^\top(p_t\eta - \theta)))$. We first calculate the gradient and Hessian of $\ell_t(\theta, \eta)$ with respect to $[\theta; \eta]$. For notation simplicity, denote $w_t := x_t^\top(p_t\eta - \theta)$.

$$\nabla\ell_t = -\left(\mathbb{1}_t \cdot \frac{s(w_t)}{S(w_t)} - (1 - \mathbb{1}_t) \cdot \frac{s(w_t)}{1 - S(w_t)}\right) \cdot \begin{bmatrix} -x_t \\ p_t \cdot x_t \end{bmatrix} \tag{25}$$

$$\nabla^2\ell_t = -\left(\mathbb{1}_t \cdot \frac{s'(w_t)S(w_t) - s(w_t)^2}{S(w_t)^2} + (1 - \mathbb{1}_t) \cdot \frac{-s'(w_t)(1 - S(w_t)) - s(w_t)^2}{(1 - S(w_t))^2}\right) \cdot \begin{bmatrix} -x_t \\ p_t \cdot x_t \end{bmatrix} [-x_t^\top, \ p_t \cdot x_t^\top]$$

$$= -\left(\mathbb{1}_t \cdot \frac{s'(w_t)S(w_t) - s(w_t)^2}{S(w_t)^2} + (1 - \mathbb{1}_t) \cdot \frac{-s'(w_t)(1 - S(w_t)) - s(w_t)^2}{(1 - S(w_t))^2}\right) \cdot \begin{bmatrix} x_t x_t^\top & -p_t \cdot x_t x_t^\top \\ -p_t \cdot x_t x_t^\top & p_t^2 \cdot x_t x_t^\top \end{bmatrix} \tag{26}$$

According to Assumption 3.8, we know that $S(w)$ and $(1 - S(w))$ are strictly log-concave, which indicates

$$\frac{d^2\log(1 - S(w))}{dw^2} = \frac{-s'(w)(1 - S(w)) - s(w)^2}{(1 - S(w))^2} < 0$$
$$\frac{d^2\log(S(w))}{dw^2} = \frac{s'(w)S(w) - s(w)^2}{S(w)^2} < 0, \forall w \in \mathbb{R}. \tag{27}$$

Since $w_t = p_t \cdot x_t^\top\eta - x_t^\top\theta$ where $p_t \in [c_1, c_2]$, we know that $w_t \in [-1, c_2]$. Since $\frac{d^2\log(S(w))}{dw^2}$ and $\frac{d^2\log(1 - S(w))}{dw^2}$ are continuous on $[-1, c_2]$, we know that

$$\mathbb{1}_t \cdot \frac{s'(w_t)S(w_t) - s(w_t)^2}{S(w_t)^2} + (1 - \mathbb{1}_t) \cdot \frac{-s'(w_t)(1 - S(w_t)) - s(w_t)^2}{(1 - S(w_t))^2}$$
$$\leq \sup_{w \in [-1, c_2]} \max\left\{\frac{s'(w_t)S(w_t) - s(w_t)^2}{S(w_t)^2}, \frac{-s'(w_t)(1 - S(w_t)) - s(w_t)^2}{(1 - S(w_t))^2}\right\} < 0. \tag{28}$$

Denote $C_l = -\sup_{w \in [-1, c_2]} \max\left\{\frac{s'(w_t)S(w_t) - s(w_t)^2}{S(w_t)^2}, \frac{-s'(w_t)(1 - S(w_t)) - s(w_t)^2}{(1 - S(w_t))^2}\right\} > 0$, and we know that

$$\nabla^2\ell_t(\theta, \eta) \succeq C_l \cdot \begin{bmatrix} x_t x_t^\top & -p_t \cdot x_t x_t^\top \\ -p_t \cdot x_t x_t^\top & p_t^2 \cdot x_t x_t^\top \end{bmatrix}. \tag{29}$$

Similarly, we know that $\frac{s(w)}{S(w)}$ and $\frac{-s(w)}{1 - S(w)}$ are continuous on $[-1, c_2]$. Therefore, we may denote $C_G = \sup_{w \in [-1, c_2]} \max\left\{|\frac{s(w)}{S(w)}|, |\frac{-s(w)}{1 - S(w)}|\right\} > 0$ and get

$$\nabla\ell_t(\theta, \eta)\nabla\ell_t(\theta, \eta)^\top \preceq C_G^2 \cdot \begin{bmatrix} -x_t \\ p_t \cdot x_t \end{bmatrix} [-x_t^\top, \ p_t \cdot x_t^\top]. \tag{30}$$

Given all these above, we have

$$\nabla^2\ell_t(\theta, \eta) \succeq C_l \cdot \begin{bmatrix} x_t x_t^\top & -p_t \cdot x_t x_t^\top \\ -p_t \cdot x_t x_t^\top & p_t^2 \cdot x_t x_t^\top \end{bmatrix}$$
$$= \frac{C_l}{C_G^2} \cdot C_G^2 \cdot \begin{bmatrix} -x_t \\ p_t \cdot x_t \end{bmatrix} [-x_t^\top, \ p_t \cdot x_t^\top] \tag{31}$$
$$\succeq \frac{C_l}{C_G^2} \cdot \nabla\ell_t(\theta, \eta)\nabla\ell_t(\theta, \eta)^\top.$$

Denote $C_e := \frac{C_l}{C_G^2}$ and we prove the lemma. ∎

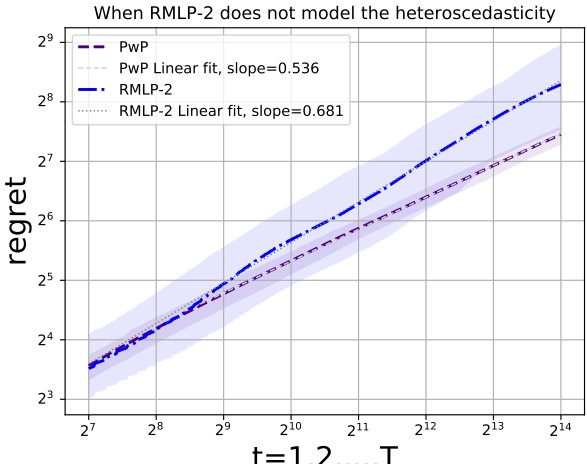

Figure 2: Regrets of PwP versus the original homoscedastic RMLP-2 algorithm. In this log-log diagram, a $O(T^\alpha)$ regret curve is shown as a straight line with slope $\alpha$. From the figure, we notice that PwP is optimal while RMLP-2 is sub-optimal, indicating the necessity of modeling homoscedasticity to achieve optimal regrets.

### A.4 PROOF OF THEOREM 4.1

*Proof.* With all lemmas above, we have

$$
\begin{aligned}
\mathbb{E}[Regret] =& \mathbb{E}[\sum_{t=1}^{T} \mathbb{E}[Reg_t(p_t)|\theta_t, \eta_t]] \\
\leq& \mathbb{E}[\sum_{t=1}^{T} C_r \cdot 2 \cdot C_J \cdot \mathbb{E}[(x_t^\top(\theta_t - \theta^*))^2 + (x_T^\top(\eta_t - \eta^*))^2|\theta_t, \eta_t] + T \cdot C_r \cdot 2 \cdot \Delta^2] \\
\leq& \mathbb{E}[\sum_{t=1}^{T} 2C_rC_J \cdot \frac{10}{C_l \cdot \Delta^2} \cdot \mathbb{E}[\ell_t(\theta_t, \eta_t) - \ell_t(\theta^*, \eta^*)|\theta_t, \eta_t] + 2C_rT\Delta^2] \\
=& \frac{20C_rC_J}{C_l\Delta^2} \mathbb{E}[\sum_{t=1}^{T} \ell_t(\theta_t, \eta_t) - \ell_t(\theta^*, \eta^*)] + 2C_rT\Delta^2 \\
=& O(\frac{d\log T}{\Delta^2} + \Delta^2 T) \\
=& O(\sqrt{dT\log T}).
\end{aligned}
$$

(32)

Here the first line is by the law of total expectation, the second line is by Lemma 4.2, the third line is by Lemma 4.3, the fourth line is by equivalent transformation, the fifth line is by Lemma 4.4, and the sixth line is by the fact that $\Delta = \min\left\{ \left(\frac{d\log T}{T}\right)^{\frac{1}{4}}, \frac{J(0,1)}{10}, \frac{1}{10} \right\}$. This holds the theorem. ∎

## B MORE EXPERIMENTS

### B.1 MODEL ADAPTIVITY

In this section, we show that it is necessary to model the heteroscedasticity. In specific, we compare PwP with the original RMLP-2 algorithm from Javanmard & Nazerzadeh (2019) that ignores heteroscedasticity in a heteroscedastic environment. We conduct both experiments for $T = 2^{14}$ rounds and repeat them for 10 epochs. The numerical results are displayed in the lower figure, plotted in log-log diagrams. From the figure, we notice that the regret of RMLP-2 is much larger than PwP. Also, the slope of regrets of RMLP-2 is $0.681 >> 0.5$, indicating that it does not guarantee a $O(\sqrt{T})$

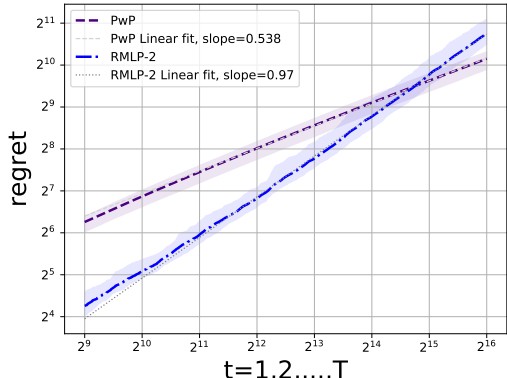

Figure 3: Regrets of misspecified PwP with expanded contexts, in comparison with a baseline RMLP-2 knowing the correct model. The results show that PwP still have a sub-linear regret in a certain period of time with context expansions, indicating that our linear demand model as Eq. (1) can be generalized to a linear valuation model as Eq. (33) in practice.

regret. In comparison, PwP still performs well as it achieves a $\sim O(T^{0.536})$ regret. This indicates that the algorithmic adaptivity of PwP to both homoscedastic and heteroscedastic environments is highly non-trivial, and a failure of capturing it would result in a substantial sub-optimality.

## B.2    MODEL MISSPECIFICATION

In Section 5, we compare the cumulative regrets of our PwP algorithm with the (modified) RMLP-2 on the *linear demand* model (as Eq. (1) or equivalently, the linear fractional valuation model as Eq. (2)). In this section, we consider a model-misspecific setting, where customer's true valuation is given by the following equation

$$y_t = x_t^\top \theta^* + x_t^\top \eta^* \cdot N_t \tag{33}$$

and the demand $D_t = \mathbb{1}_t = \mathbb{1}[p_t \le y_t]$. As a result, Eq. (33) captures a *linear valuation* model with heteroscedastic valuation.

Now, we design an experiment to show the generalizability of both our PwP algorithm and our demand model as Eq. (1). In specific, we run the PwP algorithm that still models a customer's valuation as $\tilde{y}_t = \frac{x_t^\top \tilde{\theta}^* + \tilde{N}_t}{\tilde{x}_t^\top \tilde{\eta}^*}$, where $\tilde{x}_t \in \mathbb{R}^q$ is an *expanded* version of the original context $x_t$ (i.e. $\tilde{x}_t = \pi(x_t)$ for some fixed expanding policy $\pi$) and $\tilde{\theta}^*, \tilde{\eta}^* \in \mathbb{R}^q$ are some fixed parameters[2]. Therefore, PwP is trying to learn those misspecified $\tilde{\theta}^*$ and $\tilde{\eta}^*$ although there does not exist such an underground truth.

We are curious whether the expansion of context (from $x_t$ to $\tilde{x}_t$) would leverage the hardness of model misspecification. For $x = [x_1, x_2, \ldots, x_d]^\top$, denote $x^n := [x_1^n, x_2^n, \ldots, x_d^n]^\top$. Then for any context $x \in \mathbb{R}^d$, we specify each context-expanding policy as follows:

$$
\begin{aligned}
&\pi(x; x_0, \mathbf{a}) \\
&:= [x; (x - x_0)^{a_1}; (x - x_0)^{a_2}; \ldots; (x - x_0)^{a_m}]^\top \in \mathbb{R}^{(m+1)d}.
\end{aligned}
\tag{34}
$$

The policy $\pi$ in Eq. (34) is a polynomial expansion of $x$ with index list $\mathbf{a} = [a_1, a_2, \ldots, a_m] \in \mathbb{Z}^m$, where $x_0 \in \mathbb{R}^d$ is a fixed start point of this expansion.

Now we consider the baseline to compare with. We claim that it is very challenging to solve the contextual pricing problem with customers' valuations being Eq. (33) with theoretic regret guarantees (although the $\Omega(\sqrt{T})$ lower bound given by Javanmard & Nazerzadeh (2019) still holds), and there are no existing algorithms targeting at this problem setting. However, there are still some straightforward algorithms that might approach it: For example, a max-likelihood estimate (MLE) of $\theta^*$ and $\eta^*$. In fact, we may still reuse the framework of RMLP-2 by replacing its MLE oracle according to the distribution given by Eq. (33). In the following, we will compare the performances of

---

[2]We may assume $q \ge d$ without loss of generality.

1. PwP algorithm with the misspecified linear demand model as Eq. (1), with expanded context $\{x_t\}$'s, and

2. RMLP-2 algorithm on the correct linear valuation model asEq. (33), with original context $\{x_t\}$'s.

We implement PwP and RMLP-2 on stochastic $\{x_t\}$ sequences (since RMLP-2 has already failed in the adversarial setting) and get numerical results shown as Figure 3. Here we choose $x_0 = [0.5, 0.5]^\top$ and $\mathbf{a} = [0, 1]$. For a model-misspecified online-learning algorithm, there generally exists an $O(\epsilon \cdot T)$ term in the regret rate, where $\epsilon$ is a parameter measuring the distance between the global optimal policy and the best *proper* policy (i.e. the best policy in the hypothesis set). However, our numerical results imply that PwP may still achieve a sub-linear regret within a certain time horizon $T$, whereas the baseline RMLP-2 that takes the correct model has a much worse regret. It is worth mentioning that PwP may still run into $\Omega(T)$ regret as $T$ gets sufficiently large, due to model misspecification. These results imply that

1. Our linear demand model Eq. (1) can be generalized to a linear valuation model as Eq. (33) in practice.

2. Our PwP algorithm can still perform well in model-misspecification settings, and even better than a baseline MLE algorithm with a correct model in a certain period of time.

For the first phenomenon that our demand model can be generalized with context expansion tricks, we may understand it as a Taylor expansion (and we take a linear approximation) at $x_0 = [0.5, 0.5]^\top$. For the second phenomenon that PwP outperforms RMLP-2, it might be caused by the non-convexity of the log-likelihood function of the valuation model specified in Eq. (33). As a result, while RMLP-2 is solving a non-convex MLE and getting estimates far from the true parameters, PwP instead works on an online convex optimization problem within a larger space (which probably contain the underground truth) due to context expansions. Unfortunately, we do not have a rigorous analysis of those two phenomenons.

## C  MORE DISCUSSIONS

As supplementary to Section 6, here we discuss some potential extensions and impacts of our work with more details.

**Assumption on lower-bounding elasticity as $C_\beta > 0$.**   Here we claim that the regret lower bound should have an $\Omega(\frac{1}{C_\beta})$ dependence on $C_\beta$. We prove this by contradiction. Without loss of generality, assume $C_\beta \in (0, 1)$. In specific, we construct a counter example to show it is impossible to have an $O(C_\beta^{-1+\alpha})$ regret for any $\alpha > 0$:

Firstly, let $\beta = C_\beta$. Suppose there exists an algorithm $\mathcal{A}$ that proposes a series of prices $\{p_t\}_{t=1}^T$ which achieve $O(C_\beta^{-1+\alpha})$ regret in any pricing problem instance under our assumptions.

Now, we consider another specific problem setting where $\beta = 1$ while all other quantities $\theta^*, \eta^*, \{x_t\}_{t=1}^T$ stay unchanged. Notice that the reward function has the following property:

$$r(u, \beta, p) = p \cdot S(\beta p - u) = \frac{1}{\beta} \cdot (\beta p) \cdot S(\beta p - u) = \frac{1}{\beta} \cdot r(u, 1, \beta p) \tag{35}$$

Therefore, we construct another algorithm $\mathcal{A}^*$ which proposes $C_\beta \cdot p_t$ at $t = 1, 2, \ldots, T$. According to the $O(C_\beta^{-1+\alpha})$ regret bound of $\mathcal{A}$, we know that $\mathcal{A}^*$ will suffer $C_\beta \cdot O(C_\beta^{-1+\alpha}) = O(C_\beta^\alpha)$ regret. Let $C_\beta \to 0^+$ and observe the regret of $\mathcal{A}^*$ on the latter problem setting (where $\beta = 1$). On the one hand, this is a fixed problem setting with information-theoretic lower regret bound at $\Omega(\log T)$. On the other hand, the regret will be bounded by $\lim_{C_\beta \to 0^+} O(C_\beta^\alpha) = 0$. They are contradictory to each other. Given this, we know that there does not exist such an $\alpha > 0$ such that there exists an algorithm that can achieve $O(C_\beta^{-1+\alpha})$. As a result, it is necessary to lower bound the elasticities by $C_\beta$ from 0.

**Adversarial attacks.**   Our PwP algorithm achieves near-optimal regret even for adversarial context sequences, while the baseline (modified) RMLP-2 algorithm fails in an oblivious adversary and suffer a linear regret. This is mainly caused by the fact that RMLP-2 takes a pure-exploration step at a *fixed*

time series, i.e. $t = 1, 1 + 2, 1 + 2 + 3, \ldots, \frac{k(k+1)}{2}$. This issue might be leveraged by randomizing the position of pure-exploration steps: In each Epoch $k = 1, 2, \ldots$, it may firstly sample one out of all $k$ rounds in this epoch uniformly at random, and then propose a totally random price at this specific round. However, RMLP-2 still requires $\mathbb{E}[xx^\top] \succeq c \cdot I_d$ even with this trick.

**Regret lower bounds for fixed unknown noise distributions.** We claim a $\Omega(\sqrt{dT})$ regret lower bound in Theorem 4.5 with customers' demand model being Eq. (1). However, this result does not imply a $\Omega(\sqrt{dT})$ regret lower bound for the contextual pricing problem with customers' valuation being $y_t = x_t^\top \theta^* + N_t$ adopted by Javanmard & Nazerzadeh (2019); Cohen et al. (2020); Xu & Wang (2021). This is because our problem setting is more general than theirs, and our construction of $\Omega(\sqrt{dT})$ regret lower bounds are substantially beyond the scope of this specific subproblem. So far, the best existing regret lower bound for the linear noisy model ($y_t = x_t^\top \theta^* + N_t$) is still $\Omega(\sqrt{T})$. However, we conjecture that this should also be $\Omega(\sqrt{dT})$. The hardness of proving this lower bound comes from the fact that the noises are iid over time, and it is harder to be separated into several sub-sequences across $d$ that are independent to each other.

**Algorithm and analysis for unknown link function** $S(\cdot)$**.** Unfortunately, our algorithm is unable to be generalized to the online contextual pricing problem with linear valuation and unknown noise distribution that has been studied by Fan et al. (2021). Indeed, the problem becomes substantially harder when the noise distribution is unknown to the agent. Existing works usually adopt bandits or bandit-like algorithms to tackle that problem. For example, Fan et al. (2021) approaches it with a combination of exploration-first and kernel method (or equivalently, local polynomial), Luo et al. (2021) uses a UCB-styled algorithm, and Xu & Wang (2022) adopts a discrete EXP-4 algorithm. However, none of them close the regret gap even under the homoscedastic elasticity environment as they assumed, and the known lower bound is at least $\Omega(T^{\frac{2}{3}})$, or $\Omega(T^{\frac{m+1}{2m+1}})$ for smooth ones (Wang et al., 2021). On the other hand, we study a parametric model, and it is not quite suitable for a bandit algorithm to achieve optimality in regret. In a nutshell, these two problems (known vs unknown noise distributions), although seem similar to each other, are indeed substantially different.

**Linear demand model vs linear valuation model.** In this work, we adopt a generalized linear demand model with Boolean feedback, as assumed in Eq. (1). As we have stated in Appendix B, there exists a heteroscedastic linear valuation model as Eq. (33) that also captures a customer's behavior. However, this linear valuation model is actually harder to learn, as its log-likelihood function is non-convex. It is still an open problem to determine the minimax regret of an online contextual pricing problem with a valuation model like Eq. (33).

**Ethic issues.** Since we study a dynamic pricing problem, we have to consider the social impacts that our methodologies and results could have. The major concern in pricing is *fairness*. In general, we did not enforce or quantify the fairness of our algorithm. In fact, we might not guarantee an individual fairness since PwP proposes random prices, which means even the same input $x_t$'s would lead to different output prices. Despite the perturbations $\Delta_t$ we add to the prices, the pricing model (i.e. the parameters $\theta^*$ and $\eta^*$) is updating adaptively over time. This indicates that customers arriving later would have relatively fairer prices, since the model is evolving drastically at the beginning rounds and is converging to (local) optimal after a sufficiently long time period. We claim that our PwP algorithm is still fairer than the baseline RMLP-2 algorithm we compare with, since RMLP-2 takes pure explorations at some specific time. As a result, those customers who are given a totally random price would have a either much higher or much lower expected price than those occurring in exploitation rounds. However, it is still worth mentioning that RMLP-2 satisfies individual fairness within each pure-exploitation epoch, since it does not update parameters nor adding noises then.

