# OpenReview forum: "Pricing with Contextual Elasticity and Heteroscedastic Valuation"
_ICLR.cc/2024/Conference — Submitted to ICLR 2024_

### Official Review · Reviewer_6Reb · 2023-10-29

**Soundness:** 3 good
**Presentation:** 3 good
**Contribution:** 3 good
**Rating:** 6
**Confidence:** 5

**Summary:**

This paper studies an online dynamic pricing problem by considering a novel model with feature-based price elasticity.  The authors provide a novel algorithm, ``Pricing with Perturbation (PwP)," that efficiently solves this pricing problem and obtains near-optimal regret, which matches the lower bound of regret up to log terms.

**Strengths:**

1. The presentation is clear. Beginning with the introduction part, the paper clearly lists its comparisons and generalizations from previous work. Later in the main text, the intuition of the algorithm is also well described. The assumptions made in the paper are also clearly listed and justified.

2. The novelty of the algorithm and its technical contributions are sound. The proposed Pricing with Perturbation (PwP) algorithm is smart and can efficiently solve the problem of a lack of fisher information.

3. Discussions on potential extensions of the work are discussed in detail in the appendix.

**Weaknesses:**

1. The motivation for this contextual price elasticity seems unclear.

2. Certain assumptions, such as $x^\top \eta$ having a positive lower bound, lack a real-world explanation.

3. Lack of applying this framework to real-data studies

**Questions:**

1. Can the authors present certain real-world motivations for this contextual price elasticity? e.g., why is it reasonable to rely on the context $x_t$, and is it reasonable to assume that for all $x_t$, $x_t^\top \eta$ is positive all the time?

2. About the linear assumption on $x_t^\top \eta$, can this be generalized to some non-linear function of $x_t$? Also, when $x_t$ is stochastic, can the assumption of $x_t^\top \eta>0$ be relaxed to $E[x_t^\top \eta]>0$, where $E[\cdot]$ is the expectation over $x$?

3. Can the authors provide a real-world (or semi-real) data study? on evaluating the performance of algorithms in real-life situations.

4. In terms of the presentation of simulation results, could the authors present log-log plots and compare them with the $1/2 log T$ curve? Since it would be hard to see the regret order if they are not presented in this way,

---

> ### Author Response · Authors · 2023-11-18
> **Authors Response to Reviewer 6Reb**
>
> Thank you for your positive feedback and insightful comments! Please kindly find our response as follows:
>
> 1, As in the generalized linear demand model $S(\alpha p + x_t^\top\beta)$, the real-world motivations of assuming the price elasticity $\alpha=x_t^{\top}\eta^*$ as contextual are mainly from the fact that different products have different price elasticities [Anderson et al. 1997], given how crucial they are for our daily life and work.
>
> The assumption that $\alpha=x_t^\top\eta^*>0$, i.e. a negative elasticity, is necessary for the monotonicity of a demand-price function, given that the link function $S(\cdot)=1-F(\cdot)$ is non-increasing. Stated as *the law of demand* [Gale, 1955] [Hildenbrand, 1983], that the quantity purchased varies inversely with price, is derived from the law of diminishing marginal utilities and has been widely accepted and used [Marshall, 2009].
>
> The assumption that $\exists C_{\beta}>0$ as a positive constant such that $x_t^{\top}\eta>C_{\beta}>0$ comes from the boundedness of the noise variance. Recall that an equivalent valuation model for our generalized linear demand is $y_t = \frac1{x_t^\top\eta^*}\cdot(x_t^\top\theta^*+N_t)$ where $N_t$ is an iid noise. Therefore, we know that $Var(y_t) = \frac1{(x_t^\top\eta^*)^2}Var(N_t)$. Therefore, it is necessary to assume the existence of $C_{\beta}$ in order to achieve a finite variance bound of $y_t$.
>
> We will include our explanation above and add more details in the updated version.
>
> 2, As for the generalization of our linear price elasticity model (i.e. $\alpha=x_t^\top\eta^*$), we believe that our algorithm design and analysis are still applicable within a slight generalization from Euclidean space to *known* kernel spaces. As for the generalization from $x_t^\top\eta^*>0$ to $\mathbb{E}[x_t^\top\eta^*]>0$ for stochastic setting, we do not think it is necessary given *the law of demand*, i.e. the necessity of assuming $x_t^\top>0$ listed above as our response to your first question.
>
> 3, We are actually motivated by real-world scenarios to consider a heteroscedastic setting where the price elasticity is feature-based. However, it is unfortunate that we are unable to have real-world evaluations of our algorithm, which requires either massive investments or confidential commercial-use data. On the one hand, a pricing algorithm cannot be fairly evaluated until being put into market. This is because customers’ valuations demand curves are never revealed precisely, causing a totally unknown demand on any not-proposed prices. Therefore, if an algorithm does not propose the historical price, then we cannot know how the demand would have been if this algorithm was applied. On the other hand, although big online retail companies could build up real-world demand simulators that are trained on historical sales records, we still have no access to either the simulator or the data as they are confidential. Besides, our contributions are mainly in theoretical aspects, and a real-world demand simulator is itself a challenging research topic that is out of the scope of this work.
>
> 4, It’s a good catch that a log-log plot would better show the regret rate. In fact, our plots are indeed presented in log-log diagrams, and therefore a slope-$\alpha$ line indicates an $O(T^{\alpha})$ regret.To show the regret rate, we also plotted the linear asymptote of each regret curve. The curve of $\frac12\log T$ on a logT-diagram as you suggested should be a straight line with slope $0.5$ adding some constant. This idea is brilliant, but it is not necessary since we already have such asymptotes.
>
>
> References:
>
> Marshall, A. (2009). Principles of economics: unabridged eighth edition. Cosimo, Inc..
>
> Anderson, P. L., McLellan, R. D., Overton, J. P., & Wolfram, G. L. (1997). Price elasticity of demand. McKinac Center for Public Policy. Accessed October, 13(2).
>
> Hildenbrand, W. (1983). On the" law of Demand". Econometrica: Journal of the Econometric Society, 997-1019.
>
> Gale, D. (1955). The law of supply and demand. Mathematica scandinavica, 155-169.

---

### Official Review · Reviewer_vsAQ · 2023-10-31

**Soundness:** 3 good
**Presentation:** 3 good
**Contribution:** 3 good
**Rating:** 6
**Confidence:** 4

**Summary:**

The paper investigates a context-based dynamic pricing problem, where customers decide whether to purchase a product based on its features and price. The authors adopt a novel approach to formulating customers’ expected demand by incorporating feature-based price elasticity. The paper provides a matched regret bound for the problem.

**Strengths:**

Generally speaking, from my point of view, the paper is well written. I really enjoy reading the discussions the authors make, including the relationship between two different formulations and Section 4.1.1. The technical part is solid. The idea of perturbation, though not completely novel, is quite interesting.

**Weaknesses:**

1.	In my opinion, Ban and Keskin (2021) should be given more credits. As far as I know, Ban and Keskin (2021) is the first to consider the heterogenous price elasticities which are formulated to be linear with context. At least when introducing the formulation, I think the paper should be cited and discussed more.
2.	I understand that a known link function is a good starting point and a common practice. One direction that I think might further improve the paper is to consider (or at least discuss about) an unknown link function. The reason why I mention this point is that Fan et al. (2021) studies a problem with unknown noise distribution. According to equivalence of the two formulation, it seems that it is not undoable to consider a version without knowing the link function.
3.	About the Perturbation, similar ideas can be found in the dynamic pricing literature (see, e.g., Nambiar et al. 2019). From my perspective, the only reason why the time horizon $T$ should be known in advance is because we need it to calculate $\Delta$. Nambiar et al. (2019) dynamically change the magnitude of the perturbation, which may potentially help the current algorithm to get rid of the known time horizon $T$. Please correct me if I am wrong.

Reference:
Gah-Yi Ban and N Bora Keskin. Personalized dynamic pricing with machine learning: High-dimensional features and heterogeneous elasticity. Management Science, 67(9):5549–5568, 2021.

Jianqing Fan, Yongyi Guo, and Mengxin Yu. Policy optimization using semiparametric models for dynamic pricing. arXiv preprint arXiv:2109.06368, 2021.

Mila Nambiar, David Simchi-Levi, and He Wang. Dynamic learning and pricing with model misspecification. Management Science, 65(11):4980-5000, 2019.

**Questions:**

See above.

---

> ### Author Response · Authors · 2023-11-18
> **Author Response to Reviewer vsAQ**
>
> Thank you for your support to our work. We have received some similar comments from our previous submission to NeurIPS, and we have already addressed their concerns with certain updates in our current version. Specifically, we have:
>
> 1,  We have attributed our model to Ban and Keskin (2021) who introduced the generalized linear demand model with heterogenous price elasticity (coefficient) for the first time, which well-motivates a feature-based heteroscedasticity and how it affects the price-demand relationship.  We also reduce a broadly adopted linear-valuation model to this generalized linear-demand model and show that they are equivalent.
>
> Our demand model is generally similar to theirs, but we generalize the demand model from two perspectives: (1) We assume the incoming feature sequence $[x_t]_{t=1}^T$ is adversarial while they assume iid series. (2) We assume a Boolean-censored demand while they assume an add-on iid noise on the demand. Those generalizations escalate the problem hardness substantially. As a result, their ILQX algorithm, which highly depends on the concentration of the empirical Fisher Information, shall not achieve $O(d\sqrt{T}\log T)$ in an adversarial (instead of iid) environment as we assume.
>
> Compared with our previous submission, we have added more in-depth discussion about Ban and Keskin (2021) in the current literature review. Please kindly see Sec 2.
>
> 2, Thanks for your suggestions! Unfortunately, our algorithm is unable to be generalized to the online contextual pricing problem with linear valuation and unknown noise distribution that has been studied by Fan et al. (2023). Indeed, the problem becomes substantially harder when the noise distribution is unknown to the agent. Existing works usually adopt bandits or bandit-like algorithms to tackle that problem. For example, Fan et al. (2022) approaches it with a combination of exploration-first and kernel method (or equivalently, local polynomial), Luo et al. (2023) uses a UCB-styled algorithm, and Xu et al. (2022) adopts a discrete EXP-4 algorithm. However, none of them close the regret gap even under the *homoscedastic elasticity* environment as they assumed, and the known lower bound is at least $\Omega(T^{\frac23})$, or $\Omega(T^{\frac{m+1}{2m+1}})$ for smooth ones (Wang et al. 2021). On the other hand, we study a parametric model, and it is not quite suitable for a bandit algorithm to achieve optimality in regret. In a nutshell, these two problems (known vs unknown noise distributions), although seem similar to each other, are indeed substantially different.
>
> Compared with our previous submission, we have included a detailed discussion about this unknown link function setting. Please kindly refer to Appendix C.
>
> References:
>
> Fan, Jianqing, Yongyi Guo, and Mengxin Yu. "Policy optimization using semiparametric models for dynamic pricing." Journal of the American Statistical Association (2022): 1-29.
>
> Luo, Yiyun, Will Wei Sun, and Yufeng Liu. "Distribution-free contextual dynamic pricing." Mathematics of Operations Research (2023).
>
> Xu, Jianyu, and Yu-Xiang Wang. "Towards agnostic feature-based dynamic pricing: Linear policies vs linear valuation with unknown noise." International Conference on Artificial Intelligence and Statistics. PMLR, 2022.
>
> Wang, Yining, Boxiao Chen, and David Simchi-Levi. "Multimodal dynamic pricing." Management Science 67.10 (2021): 6136-6152.
>
>
> 3, Thanks for pointing out this work and their methods. Nambiar et al. (2019) studies a contextual pricing problem with *linear demand* and is different from ours by a link function $S$, and we have included more discussions on this stream of works in our updated literature review (see Sec 2). Actually the knowledge of $T$ is not a big issue. As we indicated on the footnote of Page 3, we may always play a “doubling-epoch trick” without knowing $T$ in advance, which is introduced by Javanmard and Nazerzadeh, (2019).  In specific, we divide the whole $T$ horizon into a series of *doubling epochs*, $T_1=2, T_2=4, \ldots, T_K = 2^K$ with $K=\log_2 T$.We do not need any knowledge to make this division. Then we play our PwP algorithm in each Epoch $k$ with its time horizon $T_k=2^k$, in each of which we do a cold start independently. Therefore, the total regret is still $\tilde O(\sum_{k=1}^{\log_2 T}\sqrt{dT_k})=\tilde O(\sqrt{dT})$. This indicates we may assume a known $T$ without loss of generality.
>
> We have also corrected every typo and redundant description that we could find. Again many thanks for your detailed and professional comments! Would you please kindly consider raising the score if our updates have addressed your concerns?

---

> > ### Comment · Reviewer_vsAQ · 2023-11-22
> >
> > Thanks for all your efforts and clarifications. I really appreciated.

---

### Official Review · Reviewer_g3du · 2023-11-02

**Soundness:** 3 good
**Presentation:** 3 good
**Contribution:** 2 fair
**Rating:** 6
**Confidence:** 3

**Summary:**

This paper unifies the ``linear demand'' and the ``linear valuation'' by proposing a new demand model where each item has a feature-dependent price elasticity. The authors devise an effective online optimization algorithm that can achieve a nearly optimal regret bound. Some numerical simulations are conducted to empirically show the effectiveness of the proposed approach.

**Strengths:**

S1. A new demand model for the contextual pricing problem.

S2. The proposed algorithm has a regret bound close to the theoretical lower bound.

S3. Numerical simulations are conducted.

**Weaknesses:**

W1. Although the proposed demand model extends existing models by considering the feature-dependent price elasticity, the proposed model and online algorithm still rely on linear forms of elasticity and valuation. Remember ICLR is a deep learning conference. A potentially more suitable treatment may be substituting the linear functions with a neural tangent kernel and then devising online algorithms correspondingly.

W2. What is the major technical challenge if we replace the uniform \alpha with a feature-dependent price elasticity? The authors may want to discuss more the impact of introducing feature-dependent price elasticity terms on algorithm design as well as regret analysis.

W3. As the authors mention in Ethic issues, personalized pricing may have fairness issues. Therefore, it is essential to discuss how to deal with the cases when we add some fairness regularization terms or fairness constraints to the optimization problem.

W4. Still about personalized pricing. As the objective is purely the interest of the platform, I would like to see discussions or experimental results on how the personalized pricing algorithm affects customer well-being metrics such as consumer surplus.

**Questions:**

W2

**Details Of Ethics Concerns:**

Please refer to W3 and W4.

---

> ### Author Response · Authors · 2023-11-18
> **Authors Response to Reviewer g3du**
>
> Thanks for your detailed feedback. We will first address the technical questions, and then get to the issue you raised about pricing fairness.
>
> > W1. "ICLR is a deep learning conference" "replace linear with NTK"
>
> While ICLR starts as a deep-learning focused conference, I believe it is now widely regarded as exchangeable with ICML and NeurIPS.  The topic list in the 2024 Call for Papers seems very inclusive and we believe our paper is within scope. The AC and SAC should confirm this?
>
> As for replacing our model with a kernelized counterpart, it is potentially doable with techniques from Calandriello, Lazaric  & Valko, ICML'2017 on "Second-order kernel online convex optimization with adaptive sketching".    In short, we might be able to replace the dimension d in our regret bound with an effective dimension d_eff.  However, we emphasize that this extension would be complementary to our current work. The main new techniques that allowed us to solve the linear case would still be essential for the kernel case.  We will see if it is feasible to add this in the appendix (if it ends up being a straightforward extension).
>
> > W2. Q1. "What makes replacing a fixed price elasticity $\alpha$ by $x_t^\top\eta^*$ hard?"
>
> It actually makes learning optimal pricing policy substantially more challenging.  Note that in our setting $x_t$’s are adversarially chosen, this gives the adversary the ability to adversarially adjust the price coefficient $x_t^\top\eta^*$ which forces the learner to more actively "explore" --- hence our noise-adding procedure. In contrast, the simpler setting with an unknown but fixed alpha* still permits the same kind of implicit exploration that existing work leverages in their MLE-type methods.
>
> Existing works [e.g. Javanmard and Nazerzadeh, 2019] have studied this fixed-$\alpha$ problem setting and proposed algorithms to tackle it. Some of them achieve $O(\sqrt{T})$ regret under certain conditions, but none of them can achieve $O(\sqrt{T})$ in our contextual-elasticity and adversarial input settings.
>
> Besides, we would like to kindly remind you that we are not intentionally *introducing* feature-dependent elasticity in algorithm design and analysis. Instead, they come from the problem setting where we consider different products should have different price elasticities, depending on their features.
>
> We will include more discussions on the difference between our linear-contextual elasticity model and an unknown fixed elasticity model, in the aspects of motivations, formulations, techniques and analysis.
>
>
>
> > W3. "fairness in personalized pricing", "fairness inducing regularization"
>
> > W4. "the objective is purely the interest of the platform" "how does personalized pricing affect buyer well-being"
>
> We agree these are important problems. However, our problem setting is *not* personalized pricing!  The context $x_t$ describes only the features of the product being sold and the same price is applied to any buyer.  This is clear in the motivating example for the whole line of work on "feature-based dynamic pricing" or "contextual dynamic pricing" started in Cohen et al. 2020, and Javanmard & Nazerzadeh, 2019.
>
> We agree that our algorithm can be used for personalized pricing if $x_t$  includes features from user profiling --- that is why we are discussing this point in the first place in the paper, i.e., to warn against potential unethical applications. Also, this is not specific to our work but rather an issue of the entire body of research on contextual pricing. More fundamentally, every new technology has a potential risk of being abused. In our humble opinion, this should not be the reason why we stop research that pushes the scientific frontier. Specific to contextual pricing, our algorithm can certainly help improve market efficiency.
>
>
> References:
>
> Adel Javanmard and Hamid Nazerzadeh. Dynamic pricing in high-dimensions. The Journal of Machine Learning Research, 20(1):315–363, 2019.
>
> Cohen, M. C., Elmachtoub, A. N., & Lei, X. (2022). Price discrimination with fairness constraints. Management Science, 68(12), 8536-8552.

---

> > ### Comment · Reviewer_g3du · 2023-11-23
> >
> > Thank you for the response and clarification. I have raised my score from 5 to 6.

---

### Official Review · Reviewer_DP3X · 2023-11-03

**Soundness:** 2 fair
**Presentation:** 3 good
**Contribution:** 2 fair
**Rating:** 3
**Confidence:** 3

**Summary:**

In this work, the authors face the problem of contextual dynamic pricing in a heteroscedastic environment. The authors face this applicative problem by proposing a new theoretical framework. They provide a lower bound on the expected regret for the setting. Then, the authors provide an algorithm, for which they discuss the upper bound, which matches the lower bound up to log factors. The authors also provide a numerical validation of the solution.

**Strengths:**

The work faces a problem of interest from the applicative point of view.

The relevant literature is properly discussed.

**Weaknesses:**

The presentation can be improved, in particular from the introductory part.

The main concern is about the theoretical analysis of this paper. Indeed, an important focus of this work is related to heteroscedasticity, which is its differential part w.r.t. existing literature. However, this phenomenon is not highlighted in the analysis. For example, in Thr 4.5, the authors retrieve a bound in which such a phenomenon is not highlighted, and the result presented is already present in the literature. Furthermore, the result presented is known for a setting that is simpler than the one presented in this paper, so it holds in this scenario.

**Questions:**

See weaknesses.

---

> ### Author Response · Authors · 2023-11-17
> **Could you provide more information about your main concern?**
>
> Thanks for writing the review.
>
> However, we find some of your comments a bit hard to parse. Do you mind providing more details on what you are referring to when you wrote:
> > "this phenomenon (heteroscedasticity) is not highlighted in the analysis. "
> and that
> > "the authors retrieve a bound in which such a phenomenon is not highlighted, and the result presented is already present in the literature"
>
> Heteroscedasticity is part of our problem setup, as discussed in the paragraph right after Eqn (2) on Page 2. Theorem 4.1 provides a guarantee for our algorithm that operates in the heteroscedastic environment and shows that it has no regret against an oracle who knows both the demand and the data-dependent noise distribution.  In that sense, we did show that our method is success for in a heteroscedastic environment, didn't we?
>
> > You wrote "Furthermore, the result presented is known for a setting that is simpler than the one presented in this paper, so it holds in this scenario."
>
> Which result are you referring to?  Could you share a reference so we can address your question concretely?

---

> > ### Comment · Reviewer_DP3X · 2023-11-22
> >
> > > Heteroscedasticity is part of our problem setup, as discussed in the paragraph right after Eqn (2) on Page 2. Theorem 4.1 provides a guarantee for our algorithm that operates in the heteroscedastic environment and shows that it has no regret against an oracle who knows both the demand and the data-dependent noise distribution. In that sense, we did show that our method is success for in a heteroscedastic environment, didn't we?
> >
> > Yes, indeed the proposed algorithm is not among the concerns I raised in my review. My main concerns are related to the presentation, which should be improved in the introductory part (not clear in my opinion), and the way in which the theoretical analysis is conducted.
> >
> > > Which result are you referring to? Could you share a reference so we can address your question concretely?
> >
> > Consider the result of Thr 4.5. The theoretical analysis conducted is not formal, and cannot be verified properly given that several steps are missing. Moreover, do you agree that your setting is more complex than the one of linear bandits? If true, do you agree that the results you present is known?

---

> > > ### Author Response · Authors · 2023-11-22
> > > **Thanks for your clarification**
> > >
> > > We thank the reviewer for your clarification and elaboration on your comments.
> > >
> > > We claim that **no** inclusion relationship exists between our setting and linear bandits, and neither setting's lower bound can imply the other. There are mainly two substantial differences. On the one hand, it is our expected *demand* model (instead of the expected linear *reward* for linear bandits) that falls in a *generalized-linear* model, with a link function bounded by $[0,1]$. Consequently, the expected *reward* function in our model (i.e., the product of price and demand) is neither linear nor generalized linear. On the other hand, while each action in linear bandits is represented by a vector of dimension $d$, our action, specifically the price $p$, is a scalar. This distinction further prevents us from adopting the dependence on $d$ in the linear-bandit lower bound. to our model.
> > >
> > >
> > > Indeed, the lower bound proof in our paper is more akin to the approaches introduced by [Broder and Rusmevichientong, 2012], which also assume a Boolean-censored demand. Nevertheless, they did not account for the contextual setting in their work, resulting in an $\Omega(\sqrt T)$ regret lower bound. In our paper, we enhance their method for contextual pricing by proposing a partitioning of the time horizon based on feature dimensionality and finally get an $\Omega(\sqrt{dT})$ lower bound.

---

> ### Author Response · Authors · 2023-11-22
> **Our main contribution is the algorithm**
>
> We also would like to add that if you have a simple method to reduce linear bandits (or another problem class with known complexity) to our problem and replace our current stated lower bound, then we are happy to use that instead and drop our claim on any novel contribution about the lower bound.  As we explained in the last post, it might not be as easy as you thought.
>
> Our main contribution is to solve the problem optimally, i.e., to design an algorithm in a new problem setting with provable no-regret learning guarantees. The lower bound is a smaller contribution that we added to certify that the rate of our regret bound is optimal.
>
> Now thinking about it perhaps this was your point about how
> >“our lower bound does not highlight heteroscedasticity”.
>
> Is what you were actually trying to say that:
>
> > “adding heteroscedasticity does not seem to increase the minimax regret, therefore the problem is not interesting from the theoretical point of view”?
>
>
> Please allow us to argue for our theoretical contribution by the following table (which we could add to our introduction).
>
> |                    |Known $\alpha$ | Unknown $\alpha$ (i.i.d x) | Unknown $\alpha$ (adversarial x) | Heteroscedastic  $\alpha=x_t^T\eta^*$ |
> |:--------------------:|:----------------:|:--------------------------:|:--------------------------------:|:---------------------------------------:|
> | Regret upper bound | $d \log T$     | $\sqrt{d T}$  |   $\infty \Rightarrow \sqrt{dT}$ **(new)**          |  $\infty \Rightarrow \sqrt{dT}$ **(new)**                 |
> | Regret lower bound | $d\log T$       | $\sqrt{T}$               | $\sqrt{T}$                     | $\sqrt{T} \Rightarrow \sqrt{dT}$ **(new)**                 |
>
> So our theoretical contribution is to settle the problem for the column on the right by proving a new upper bound (and slightly improving the lower bound).
>
> While the heteroscedastic case might not be *information-theoretically harder* than the homoscedastic case with unknown $\alpha$, that doesn't mean that it is easy to construct an algorithm. To say it differently, our results imply that we can solve the harder problem with heteroscedasticity with the same asymptotic regret bound compared to an easier problem.  That is a good news, isn’t it?

---

### Meta-Review · Area_Chair_t8cz · 2023-12-12

**Metareview:**

This paper innovatively tackles online contextual dynamic pricing, introducing a unique approach with feature-based price elasticity and proposing the "Pricing with Perturbation (PwP)" algorithm. With optimal regret bounds and insights into the interplay between contextual elasticity and heteroscedastic valuation, it offers a valuable contribution to the field.

The reviewers also agree that the paper has the potential to be a competitive work but the current presentation of the theoretical claims is not formal, e.g. in Theorem 4.5. The lower bound is performed without any formal criteria. Informal theoretical statements can be misleading and it does not sound like this would be a convincing contribution to the community in the present state.

We strongly encourage the authors to resubmit the paper after substantiating their theoretical claims on a solid formal ground, as well as incorporating the other suggestions from the reviews.

**Justification For Why Not Higher Score:**

N/A

**Justification For Why Not Lower Score:**

N/A

---

### Decision · Program_Chairs · 2024-01-16

Reject